# Elevated CO_2_ and Reactive Oxygen Species in Stomatal Closure

**DOI:** 10.3390/plants10020410

**Published:** 2021-02-23

**Authors:** Xiaonan Ma, Ling Bai

**Affiliations:** State Key Laboratory of Crop Stress Adaptation and Improvement, School of Life Sciences, Henan University, Kaifeng 475004, China; maxiaonan@henu.edu.cn

**Keywords:** CO_2_, ROS, stomatal movement

## Abstract

Plant guard cell is essential for photosynthesis and transpiration. The aperture of stomata is sensitive to various environment factors. Carbon dioxide (CO_2_) is an important regulator of stomatal movement, and its signaling includes the perception, transduction and gene expression. The intersections with many other signal transduction pathways make the regulation of CO_2_ more complex. High levels of CO_2_ trigger stomata closure, and reactive oxygen species (ROS) as the key component has been demonstrated function in this regulation. Additional research is required to understand the underlying molecular mechanisms, especially for the detailed signal factors related with ROS in this response. This review focuses on *Arabidopsis* stomatal closure induced by high-level CO_2_, and summarizes current knowledge of the role of ROS involved in this process.

## 1. Introduction

Stomata, the tiny pores formed by pairs of guard cells on the surfaces of plant leaves, are the main portals for gas exchange, water transpiration and pathogen invasion in plants. Plants adapt to changes in the external environment by regulating the aperture of stomata. The movement of guard cells is regulated by a variety of environmental factors, including water status, light, carbon dioxide (CO_2_) concentration and pathogen attack, as well as internal signals such as phytohormones, calcium and reactive oxygen species [1,2].

CO_2_, as an important regulator of stomatal movement and development, greatly affects the growth and biomass of plants. With the development of modern industry and the increase in human activities, the CO_2_ content in the atmosphere is increasing year by year. In January 2021, data from the Mauna Loa Observatory show that the atmospheric background CO_2_ content will exceed 417 parts per million (ppm) (https://www.metoffice.gov.uk/research/climate/seasonal-to-decadal/long-range/forecasts/co2-forecast (accessed on 22 February 2021)). The continuous rise of CO2 in the atmosphere resulted in exponential increasing of intercellular CO_2_ levels in leaf cells [3,4,5]. The profound effect of CO_2_ on stomatal development has been confirmed using plant fossils with CO_2_ [6,7]. In addition to changes in development, the intercellular CO_2_ concentration also influences stomatal responses [8,9]. Although CO_2_ signaling is well established to regulate stomata movement, its molecular mechanisms are not fully understood.

In stomata, the CO_2_ signaling pathway is complex. It is generally thought that during stomatal closure induced by elevated CO_2_ concentration, CO_2_ is first converted into bicarbonate (HCO_3_^−^) by beta Carbonic Anhydrase 1 (βCA1) and beta Carbonic Anhydrase 4 (βCA4) [10]. HCO_3_^−^ is perceived and transported by the antimicrobial extrusion (MATE) type transporter RESISTANCE TO HIGH CO_2_ (RHC1) [11]. Subsequently, Mitogen-Activate Protein Kinase 4 (MPK4) and MPK12 inhibit HIGH LEAF TEMPERATURE 1 (HT1), which then activates OPEN STOMATA 1 (OST1). Activation of OST1 facilitates the S-type anion channel to mediate anion effluxes, resulting in stomatal closure [10,11,12,13,14,15,16,17].

Reactive oxygen species (ROS) are major regulators of stomatal movement, particularly in response to abiotic and biotic stress [18,19]. ROS, particularly hydrogen peroxide (H_2_O_2_) and superoxide (O_2_^−^), are widely produced within different cellular compartments of plants. The evolution and maintenance of different sources for ROS production is most likely due to the requirement for intricate control of oxidative signaling, given that ROS can be cytotoxic and mutagenic and, for their proper function in signaling, their production must be tightly regulated both temporally and spatially [20]. Recent progress for the high-level CO_2_-induced stomatal closure showed that ROS function as the key factor [21]. However, components and mechanism underlying this ROS mediated CO_2_ signaling pathway are still not so clear, with only a few factors have been disclosed. This review summarizes on the process of high-level CO_2_-induced closure in *Arabidopsis* and focuses more on current knowledge related with ROS signal. Understanding of how ROS signal have been engaged in high-level CO_2_-induced stomatal movement is essential for CO_2_ signaling, which will demand extensive research.

## 2. Importance of CO_2_ Regulation of Stomatal Conductance

Under dark conditions, respiration leads to a rapid increase in intercellular CO_2_ concentration ([CO_2_]_cys_) in plant leaves, exceeding 600 ppm [22]. However, under light conditions, [CO_2_]_cys_ drops below 200 ppm [22]. High environmental CO_2_ concentration induces stomatal closure, whereas low environmental CO_2_ concentration triggers stomatal opening. Despite its importance, little is known about the molecular mechanisms underlying the initial response of stomata to CO_2_.

CO_2_ negatively affects stomatal conductance, or the rate of passage of materials through stomata, by reducing both the number of stomata per unit of leaf area and the stomatal apertures. Decreases in stomatal conductance lead to decreases in transpiration, reducing the loss of water from leaves, which is beneficial to plant water conservation [23]. In the case of relatively high CO_2_ levels, fewer stomata and smaller apertures reduce the heat evapotranspirative cooling capacity of leaves, which aggravates heat stress under water shortage [24,25]. Notably, heat stress greatly reduces plant resistance and crop yields globally, affecting agricultural production and possibly nutrient content [26,27]. By contrast, greater stomatal conductance improves crop yield [28,29,30]. Therefore, under conditions of sufficient water, higher CO_2_ concentrations reduce stomatal conductance, which may contribute to suboptimal yields [25].

In some species, CO_2_ concentration also affects leaf development and leaf area. These responses directly affect leaf water conservation capacity and plant biomass [25]. Future research is needed to explore the molecular mechanism by which CO_2_ regulates plant gas exchange.

## 3. Mechanism of High CO_2_-Induced Stomatal Closure

Although our understanding of plant responses to environmental CO_2_ has advanced, the mechanisms underlying CO_2_ signal perception and transduction are still not clear. After entering the cell, CO_2_ is sensed by receptors on the plasma membrane. In animals, CO_2_ perception is believed to be a vestigial sensory mechanism [31] associated with carbonic anhydrase [32]. Adenylyl cyclase enzymes play important roles in CO_2_ sensing in mammalian and fungal systems [31]. Plant homologs of α- or γ- carbonic anhydrases have not been identified, while β-carbonic anhydrases have been characterized [10,33]. β-carbonic anhydrases have been proposed to carry out the CO_2_ receptor function in plants [31], which is important in the stomatal movement response to changes in CO_2_ concentration in *Arabidopsis*. Of the six *Arabidopsis thaliana* β-carbonic anhydrases, βCA4 is localized at the plasma membrane, while βCA1 is mainly targeted to chloroplasts when transiently expressed in *Nicotiana benthamiana* cells [10,34]. βCA1 and βCA4 convert CO_2_ into HCO_3_^−^, and both function in CO_2_-induced stomatal movement. The expression of either βCA4 at the plasma membrane or βCA1 in chloroplasts in guard cells of *ca1 ca4* double-mutant plants restored CO_2_-induced stomatal closure [35]. The expression of animal carbonic anhydrase in the double mutant (*ca1 ca4*) similarly restored CO_2_-induced stomatal closure [10,35]. These data suggest that the catalytic activity of carbonic anhydrase is involved in the signaling pathway of CO_2_ stimulation of closure, and HCO_3_^−^ and/or H^+^ may act as second messengers in this signal transduction in plants [10]. The apparent conservation of carbonic anhydrase activity between plants and animals implies that they sense CO_2_ by similar mechanisms.

Protein HT1 has been characterized as an important factor in CO_2_ signaling. As the first CO_2_-response mutant, *ht1* has been identified by thermal imaging, which has higher leaf temperature and reduced stomatal CO_2_ response [12]. *HT1* encodes a putative MAPKKK kinase that negatively regulates stomatal responses to changes in CO_2_ levels, but does not affect abscisic acid (ABA) signaling [12], suggesting that there is an ABA-independent signaling pathway in CO_2_-regulated stomatal response. *ca1 ca4 ht1-2* triple mutants exhibit a similar CO_2_ response as *ht1-2*, suggesting that HT1 acts downstream of βCA1 and βCA4 in CO_2_-induced stomatal movement [10]. βCA4 interacts with PIP2;1, which might transport apoplastic CO_2_ [36]. The OST1 protein kinase is required for CO_2_-induced stomatal closing [37]. A recent study proposed that HT1 deactivates OST1 [11]. However, another study found HT1 to be epistatic to OST1 in a high-CO_2_-induced stomatal closing assay [38]. Since the function of OST1 in stomatal closure is mainly through regulating ion channels, the research of anion channels in response to high levels of CO_2_ is helpful for establishing a comprehensive signal pathway.

The role of HCO_3_^−^ in regulating stomatal movement has been studied using patch clamp experiments. Stomatal closure requires the activation of anion channels in guard cells. Increasing the concentration of HCO_3_^−^ in the cytoplasm activates anion channels [37]. The MATE transporter protein RHC1 is a receptor for HCO_3_^−^ [11]. Increased CO_2_ concentration enhances anion channel activity in guard cells [10]. S-type anion channels in the plasma membrane of guard cells might provide a central control mechanism for stomatal closing [39]. The S-type anion channel SLAC1 plays a crucial role in stomatal movement during signaling [40]. SLAC1 is activated by elevated intracellular HCO_3_^−^ levels in guard cells [37]. Nevertheless, the N terminus is important for ABA signaling, and the SLAC1 transmembrane domain responds to CO_2_ but not the N terminus or C terminus [41]. Recent studies have shown that residue R256 of SLAC1 is required for elevated CO_2_ induced stomatal closure, but not for ABA-induced stomatal closing [42]. Both OST1 and GUARD CELL HYDROGEN PEROXIDE-RESISTANT1 (GHR1) activate SLAC1 in oocytes [43,44]; this activation is inhibited by HT1 [14]. MPK12 counteracts this inhibition [14]. In a recent natural variation study, MPK12 was also identified as a key player in the *Arabidopsis* Cvi-0 accession for guard cell CO_2_ signaling [15]. These findings suggest that MPK12 inhibits HT1, and HT1 inhibits OST/GHR1-activated SLAC1. Unlike HT1, which mainly functions in CO_2_ signaling, MPK12 is also activated by ABA or H_2_O_2_ [45]. Co-expressing βCA4 and PIP2;1 with OST1-SLAC1 or CPK6/23-SLAC1 in oocytes activates SLAC1 via extracellular CO_2_ [36]. These findings suggest that in the presence of various protein kinases, SLAC1 can be regulated by cytosolic CO_2_/HCO_3_^−^ that has been transported by PIP2;1 and converted by βCA4 in oocytes. Till now, a simple regulation pathway for CO_2_-induced stomatal closure has been disclosed.

Since stomatal movement is always associated with oscillations of cytosolic [Ca^2+^]_cy__t_ [46], research on the stomata response to high levels of CO_2_ [47] has also identified the involvement of calcium ions in *Commelina communis* [48,49] and *Arabidopsis thaliana* [2,50]. This conclusion is based on experiments in which Ca^2+^ accumulated in guard cells subjected to elevated CO_2_ concentration, and high CO_2_-induced stomatal closure was impaired in the presence of Ca^2+^ chelators, such as 1,2-Bis(2-Aminophenoxy)ethane-N,N,N′,N′-tetraacetic acid (BAPTA) or ethylenedinitrilotetraacetic acid (EDTA) [48,49,50]. Exposure to reduced CO_2_ concentration triggered more [Ca^2+^]_cyt_ transients in guard cells than did exposure to elevated CO_2_ concentration [2]. As guard cells produce “spontaneous” cytoplasmic Ca^2+^ transients, and Ca^2+^ is required for high CO_2_-induced stomatal closure, elevated CO_2_ concentration might enhance the sensitivity of stomatal closing mechanisms to [Ca^2+^]_c__yt_ [51]. In agreement with this hypothesis, CO_2_-derived bicarbonate enhances the Ca^2+^ sensitivity of the S-type anion channel activation in guard cells [37]. The increase in [Ca^2+^]_c__yt_ transients in response to higher CO_2_ levels implicates this second messenger in the complex mechanism underlying CO_2_-regulated stomatal movement, an idea that may need more extensive research.

Although an ABA-independent pathway has been demonstrated in CO_2_-regulated stomatal response, ABA signaling components have also been implicated, suggesting that a shared regulatory molecular mechanism may involve ABA and CO_2_ in stomatal movement. At both low and high CO_2_ concentrations, the [Ca^2+^]_c__yt_ of the ABA-insensitive mutant *gca2* changes at a similar rate, and the mutant exhibits strongly attenuated stomatal closure in response to increased CO_2_ in leaves [2]. The *gca2* mutant also exhibits an altered ABA-induced [Ca^2+^]_cyt_ pattern in guard cells [46], suggesting that Ca^2+^ alteration mediated by GCA2 (GROWTH CONTROLLED BY ABSCISSIC ACID 2) functions downstream or at the convergence point of CO_2_ and ABA signaling. Therefore, the same factors as in ABA signaling may have also been implicated in CO_2_-induced stomatal closure, suggesting that the same mechanism exists for stomatal closure, which may be helpful for understanding CO_2_ signaling.

## 4. ROS in Stomatal Closure

Stomatal closure is accompanied by increased ROS level in the guard cell apoplast and chloroplasts in response to various treatments [19,52]. In *Arabidopsis*, apoplastic ROS are mainly produced by plasma membrane-localized NADPH oxidases (RESPIRATORY BURST OXIDASE HOMOLOGS, RBOHs) and cell-wall peroxidases [19,53,54], whose activities are strongly inhibited by diphenylene iodonium (DPI) and salicylhydroxamic acid (SHAM), respectively [55,56,57,58,59].

In *Arabidopsis* guard cells, there are two main isoforms of NADPH oxidases, AtRBOHF and AtRBOHD, which are regulated by ABA-dependent processes [60]. ABA-triggered stomatal response is significantly reduced in the *atrbohF* mutant, a phenotype that is enhanced in the *atrbohD atrbohF* double mutant but absent in the *atrbohD* single mutant [60]. Due to its obvious role in pathogen-triggered ROS burst, RBOHD is more commonly recognized to function in plant immune defense [61] and might therefore play a role in stomatal movement. Both NADPH oxidases have been demonstrated to function in guard cell CO_2_ responses by staining *atrbohD atrbohF* with H2DCFDA, and the CO_2_-induced ROS burst requires ABA signaling [21], which implies that the two oxidases are essential for both ABA and CO_2_-induced stomatal response.

Apoplastic ROS are also produced by other oxidases [62]. For example, in *Vicia faba* and *Arabidopsis thaliana*, copper amine oxidase and polyamine oxidases are involved in the production of the ROS in the process of ABA- and ethylene-induced stomatal closure [63,64]. The involvement of these oxidases in apoplastic ROS production comes from inhibitor studies, and further research is needed to understand their specific and detailed function and significance in ROS-induced stomatal regulation.

Apoplastic ROS production initiates the activation of plasma membrane Ca^2+^ channels, leading to an increase in [Ca^2+^]_c__yt_ levels [51]. The molecular identity of these inducible plasma membrane Ca^2+^ channels is not clear. In the cytosol, Ca^2+^ stimulates the activation of NADPH oxidases either directly by binding to their cytoplasmic EF hands (the EF hand is a helix-loop-helix structural domain or motif found in a large family of calcium-binding proteins) [65] or indirectly by affecting their phosphorylation by CALCIUM DEPENDENT PROTEIN KINASES (CPKs) [66]. Upon Ca^2+^ binding, CALCINEURIN-B LIKE PROTEINS (CBLs) interact with CPKs and CBL-interacting PROTEIN KINASES (CIPKs) [66], and a particular complex CBL1/CBL9-CIPK26 is formed, which then phosphorylates and activates RBOHF [67]. The increase in [Ca^2+^]_cyt_ is also sensed in the chloroplasts, where a thylakoid membrane-associated Ca^2+^-binding protein, CALCIUM SENSING RECEPTOR (CAS), is activated through an unidentified mechanism [51]. The activation of CAS is responsible for a chloroplast ROS burst and the release of Ca^2+^ from thylakoids [68,69,70,71], both of which contribute to cytoplasmic Ca^2+^ oscillations, apoplastic Ca^2+^-induced stomatal closure during plant immune defense [69]. The drought sensitivity of the *Arabidopsis cas* mutant is caused by the improper closure of stomata [72], which highlights the importance of chloroplastic Ca^2+^ signaling in stomatal regulation [51].

Although there is clear evidence for the involvement of ROS in the regulation of stomatal aperture, it is not known how the ROS signals are sensed in the guard cell apoplast. Identification of the ROS and redox sensors is a major challenge in plant ROS research. In guard cells, only a few ROS sensing mechanisms are involved in stomatal regulation: redox regulation of the GHR1 apoplastic domain [44], of OST1 [73] and of CPK1 [74]. GHR1 is proposed to be involved in the perception of apoplastic ROS, a plasma membrane-associated atypical leucine-rich repeat receptor-like protein kinase. The apoplastic C-terminal domain of GHR1 has two conserved cysteines (C-57 and C-66) that are necessary for the correct function of the protein [44]. GHR1, as an inactive pseudokinase that mediates the activation of SLAC1 via interacting with CALCIUM-DEPENDENT PROTEIN KINASE 3 (CDPK3), could potentially act in stomatal closure as a scaffolding component [75]. Therefore, GHR1 is implicated as a central regulator of guard cell CO_2_ and early ABA responses for stomatal movement by mediating ROS signaling. However, more detailed research is demanded to elucidate the mechanism of ROS in stomatal movement induced by different stimuli.

## 5. Function of ROS Signaling in CO_2_-Induced Stomatal Closure

Despite their importance, not much effort has been dedicated to understanding the function of ROS in CO_2_-regulated stomatal behavior. Treatment with high concentrations of HCO_3_^−^ induces ROS production and promotes plasma membrane-localized NADPH oxidase-dependent stomatal closure [76], suggesting that ROS play a role in CO_2_-induced stomatal closure.

Chater et al. concluded that ROS are required for stomatal closure induced by high CO_2_ concentrations, as no stomatal closure was observed in *rbohD rbohF* double mutants [21,77], but there was a strong ROS decrease in guard cells at elevated CO_2_ concentrations. These observations could be explained by a reduction in the oxygenase activity of RuBisCO at high CO_2_ concentration, and hence a reduction in ROS production by glycolate oxidase activity linked to photorespiration [78]. These findings suggest that ROS generated by NADPH oxidase play an important role in high-CO_2_-induced decreases in stomatal aperture.

Cell wall-bound SHAM-sensitive peroxidases also take part in apoplastic ROS production around guard cells [57,58]. Two cell-wall peroxidase-encoding genes, *PRX33* and *PRX34*, which are highly and preferentially expressed in guard cells, are also involved in ROS production in CO_2_-induced stomatal closure [77]. Pharmacological and genetic studies show that ROS generated by both NADPH oxidases and cell-wall peroxidases contribute to high-CO_2_-induced stomatal closure. The high-CO_2_-induced reduction in stomatal aperture is efficiently abolished by either the NADHP oxidase inhibitor DPI or the cell-wall peroxidase inhibitor SHAM, suggesting that cell-wall peroxidases function in high-CO_2_-induced stomatal closure. Similar to *rbohD rbohF* double mutants, stomatal apertures of both *prx33-3* and *prx34-2* mutant lines fail to close in response to high CO_2_. ROS accumulation is not triggered by high CO_2_ in *prx33-3 prx34-2* or *rbohD rbohF* mutants, in marked contrast to the >50% ROS increase in wild type. These results indicate that in addition to RBOHD and RBOHF, cell-wall peroxidases including PRX33 and PRX34 play an essential role in high-CO_2_-induced ROS generation [77]. However, it is not certain whether other oxidase proteins also participate in CO_2_-induced ROS accumulation, and more high-CO_2_-induced ROS generation related proteins need to be identified.

Proteins that regulate ROS homeostasis in CO_2_-induced stomatal closure have also been screened. High-CO_2_-induced ROS production and stomatal closure in *big* mutants are compromised [77]. The BIG/CIS1 protein is involved in diverse processes, including auxin transport, light and hormonal signaling, vesicle trafficking, endocytosis, phosphate deficiency tolerance, and the dynamic adjustment of circadian period [79,80,81,82,83,84,85,86]. He et al. found that *cis1* mutants are compromised in both elevated CO_2_-induced closure and reduction in stomatal density [87]. *cis1* mutants have a significantly lower leaf surface temperature compared with the wild type when exposed for 40 min to 1500 ppm CO_2_. S-type anion channel activity is disrupted by HCO_3_^−^ treatment compared with the wild type [87]. This work suggests that the signaling pathway for ROS in CO_2_-regulated stomatal movement may include other signals, and unraveling the control mechanism will depend on identifying more of these molecules.

## 6. ROS Are the Nodes of CO_2_ and ABA Signaling during Stomatal Movement

Since both ABA- and high CO_2_-induced stomatal closure involve the activation of SLAC1 in guard cells, there may be a convergence between ABA and CO_2_ signaling. Several mutations causing stomatal ABA insensitivity, such as *abi1-1*, *abi2-1* [88,89], *ost1* and *ghr1* [14,88], impair stomatal responses to high CO_2_ concentrations, indicating that there is a relationship between ABA and CO_2_. Stomata of the *gca2* mutant are insensitive to both ABA and high CO_2_ signaling [2,46]. ABA receptors, PYR/PYL/RCARs, are also involved in CO_2_ signaling, as the inactivation of several of these proteins impairs stomatal closure in elevated CO_2_ concentrations [21,88], warranting further identification of the role of these receptors in CO_2_ signaling.

Although several key components of ABA signaling are connected with stomatal responses to high CO_2_ concentration, suggesting that an ABA-dependent pathway participates in CO_2_ signaling, ABA-independent components also exist. ABA-induced stomatal closure is completely functional in the mutants of HT1 and MPK12, whereas these plants are deficient in CO_2_-induced stomatal movements [14,15]. Moreover, experiments which aimed to dissect which parts of the SLAC1 anion channel are important for ABA- and for high-CO_2_-induced stomatal closure showed that transgenic plants expressing the SLAC1 anion channel without both C- and N-terminal regions still respond to changes in CO_2_ concentration, but remain ABA-insensitive. Thus, the ABA-induced activation of SLAC1 appears to involve its C- and N-terminal regions, whereas high-CO_2_-induced stomatal closure appears to rely only on the transmembrane region [41].

Current knowledge of both ABA- and high-CO_2_-induced stomatal closure suggests at least three partially overlapping pathways: (1) the direct perception of HCO_3_^−^ by SLAC1 in the presence of protein kinases that activate anion channel SLAC1 [36]; (2) the ABA-independent pathway; and (3) the ABA-dependent component that partially mediates high-CO_2_-induced stomatal closure [21,37,88]. ABA signaling that activates OST1 and CPKs by the suppression of PP2Cs could enhance SLAC1 sensitivity to HCO_3_^−^, and directly trigger SLAC1 anion currents, although this hypothesis needs to be verified in future research. When the CO_2_ supply for plants is sufficient, it is possible that plants under water stress react to increased [CO_2_]_cyt_ faster and stronger than plants with satisfactory water supply, thus saving water in leaves. This could explain the importance of ABA signaling for CO_2_-induced stomatal movements, which would enable plants to adapt to changing environmental conditions. These reports suggest that ABA and CO_2_ signaling pathways in stomatal closure are not totally the same, but special factors or regulation systems are recruited.

The overlap of CO_2_ and ABA signaling also indicates that ROS production in guard cells can increase in response to high CO_2_ concentration, similar to ABA-induced stomatal closure [51]. We previously showed that ROS accumulate in guard cells treated with HCO_3_^−^ or high CO_2_ [21,76,90]. A connection between CO_2_ and ABA signaling is supported by the absence of ROS accumulation in stomata under elevated CO_2_ concentration in the ABA-deficient double mutant *nced3 nced5*, the triple *pyr1 pyl1 pyl4* and quadruple *pyr1 pyl1 pyl2 pyl4* mutants [21]. Similar to ABA, elevated CO_2_ induces ROS formation by NADPH oxidases [21,90]. In *rbohD rbohF* double mutants, guard cells are insensitive to CO_2_/HCO_3_^−^. These mutants also fail to produce ROS in guard cells in response to elevated CO_2_ [21,76]. The impaired accumulation of ROS in guard cells and decreased stomatal closure in response to elevated CO_2_ concentration are also observed in the tomato mutant *rboh1* [90]. Since ROS-related signal components in ABA-induced stomatal closure have been revealed, the involvement of these ABA factors in ROS signaling in response to CO_2_ suggests that high-CO_2_ and ABA signaling may converge at ROS, and similar downstream components may function in both pathways.

## 7. Concluding Remarks and Future Directions

This review summarizes the molecular mechanisms of the ROS signaling network in plant stomatal movement, primarily under high CO_2_ concentrations (Figure 1). Elevated [CO_2_]-mediated stomatal closure requires an increase in ROS. However, great difficulty still exists in understanding the complex interactions within the guard cell signaling networks in response to changes in CO_2_, which include ROS signaling. Recent research has highlighted the complex interplay between CO_2_, redox/ROS signaling and phytohormonal regulation in the control of stomatal movements, and as discussed in this review, there is cross-talk between ABA and CO_2_ signaling by ROS during stomatal closure. The importance of guard cells in photosynthesis and transpiration has attracted considerable effort toward understanding how guard cells perceive and transmit signals from the environment. The identification of proteins that can sense and transmit changes in CO_2_/HCO_3_^−^ and molecules that function in regulating ROS signals in guard cells is urgent to improve understanding of the regulation of stomatal movement in response to CO_2_.

Future breakthroughs will most likely come from the development of new methods and technologies that enable real-time imaging of physiological indexes such as cellular localization of ROS, Ca^2+^, and pH in response to various stimuli including CO_2_; obtaining novel genetic mutants to identify proteins involved in CO_2_ response will also be important [47]. Furthermore, plant altered DNA methylation has been implicated in various environmental stress response [91], increasing evidences demonstrated that multiple epigenetic factors and their interactions with hormones, especially auxin will be another direction for understanding high CO_2_-induced stomatal closure [92]. The isolation of methylation-related mutants will be a useful tool for exploring molecular mechanisms. Future research should also seek to understand the complex interactions between various guard cell signaling pathways and how guard cells modulate the interactions among hormones, ROS, CO_2_ and Ca^2+^ homeostasis. In addition, the translation of such knowledge from model plants to important crop species will be increasingly important for improving stress resistance and increasing yield.

CO_2_ has been transported by PIP2;1 and converted into HCO_3_^−^ by βCA4, which is located at the plasma membrane and interacts with PIP2;1. The accumulation of cytosolic HCO_3_^−^ leads to the suppression of HT1 by MPK4/12 and RHC1. Repressed HT1 inhibits OST1, resulting in the activation of SLAC1. OST1 phosphorylates respiratory burst ROBHs to produce ROS and induces stomatal closure. GHR1 is an inactive pseudokinase that mediates the activation of SLAC1. BIG1 activates SLAC1 when cytosolic HCO_3_^−^ accumulates and produces ROS that induce stomatal closure.

PRXs, peroxidases; RBOHs, respiratory burst oxidase homologs; PIP2;1, plasma membrane intrinsic protein 2;1; βCA4, beta carbonic anhydrase 4; RHC1, resistance to high CO_2_;HT1, high leaf temperature 1; MPK4/12, protein kinase 4/12; OST1, open stomata 1; ABI1, ABA insensitive 1; BIG1, CO_2_ insensitive 1 GHR1, guard cell hydrogen peroxide-resistant 1; SLAC1, slow anion channel-associated 1; ROS, reactive oxygen species; P, phosphorylation.

## Figures and Tables

**Figure 1 plants-10-00410-f001:**
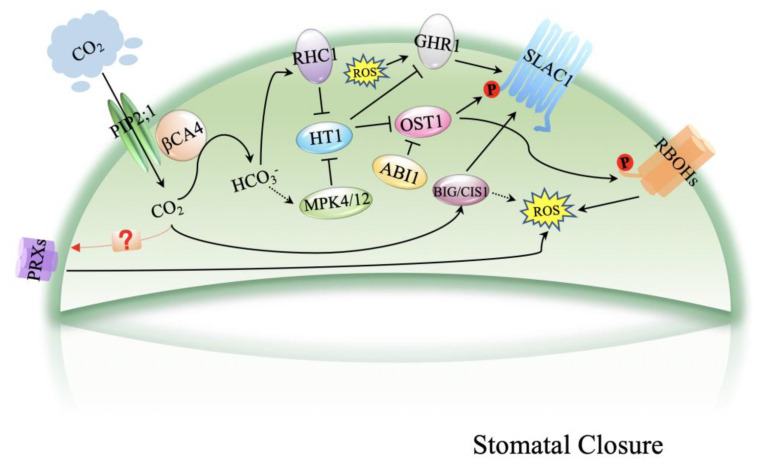
Schematic model of CO_2_-induced stomatal closure.

## Data Availability

Data sharing is not applicable.

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
