# Peer review of "Elevated CO2 and Reactive Oxygen Species in Stomatal Closure"

_plants, 2021, doi:10.3390/plants10020410_

Round 1

Reviewer 1 Report

In their manuscript, Ma and Bai reviewed the literature about the implication of ROS in CO2-induced stomatal movements. The present review is organized, well written, and of high interest. Furthermore, the cited literature is thorough and up to date.

I have some small comments that could help improve the present manuscript:

  1. In the introduction (line 43), the authors misspelled HT1 (HIGH LEAF TEMPERATURE 1).
  2. When mentioning blue light in their text, the authors failed to mention a recent publication (Hiyama et al., Nature Comm 2017) describing the role of CBC1 and CBC2, two kinases implicated in both blue light- and CO2-induced stomatal movements through the direct regulation of HT1.
  3. I also noticed that the authors failed to cite another recent piece that described the direct role of bicarbonate in SLAC1 and other S-type anion channels regulation (Zhang et al., PNAS 2018).
  4. While the authors described the role of GHR1 thoroughly, I realized that they did not cite a recent paper showing that the pseudo-kinase interacts with CDPK3, and that GHR1 could potentially act in stomatal closure as a scaffolding component (Sierla et al., The Plant Cell 2018).
  5. The authors also forgot to give the full name of GHR1 (GUARD CELL HYDROGEN PEROXIDE-RESISTANT1) in the manuscript.
  6. Finally, I think it would be important to integrate GHR1 in Figure 1 as the pseudo-kinase is a crucial component bridging the gap of both ROS and CO2 signaling.

Reviewer 2 Report

General comment: in the abstract, you mention the main focus on the ROS, while ROS starts only at line 168 and represents less than half of the review.

he second point that there is only two papers from 202 and one from 2019. The majority of the citations are from 2012-2015. How can we tell about summarizing the current state of arts in this case?   Abstract:

Sentences are too long and complicated. Please, make it more simple and more understandable.

Line 15: „focuses on the role of ROS in Arabidopsis“- not clear point: role in what? Do you mean in stomata function or in general?

Line 29: „April 2014“, but now is 2021. Please, try to find more recent data.

The sentences on lines 30-33 are similar. Please, combine it.

Line 36:  „CO2 signaling is well established to regulate stomata“ do you mean stomata function?

Lines 85-91: please, make text more logic. So far it is rather mixing of statements without logical connection.

Line 106: „High Leaf Temperature 1 (HT1)“ – please, mention that this is a gene.

Line 152: ROS formation or ROS accumulation?

Lines 175-193: please, formulate more clear conclusion about link of NADPH oxidase/peroxidase and CO2.

Lines 227-231: please, clarify which ROS do you mean: NADPH oxidase produced superoxide, not H2O2.

Lines 240-242: I would suggest not to mix stomata density what regulated by hormonal signaling, cell polarity, assymmetric cell division with stomata closure.

Line 184: you describe cell-wall peroxidase and again cell-wall peroxidase appeared on line 243. Please, make it more logical, without jumping. It will be logical to mcombine lines 184-186 with 243- 255.

Line 296: left bracket is missing.

Lines 320-321: „The elevated [CO2]-mediated control of stomatal density and aperture requires an increase in ROS“ – there is no any evidences that ROS is require for stomatal density. Please, also clarify which ROS do you mena: NADPH oxidase produced superoxide, but peroxidase- H2O2.  

Please, add epigenetic part to the future direction.

Round 2

Reviewer 2 Report

Thanks, all comments have been answered. 

Only very minor points:

Line 29-31: please, edit: provide a link to at least webpage, and remove one 2021.

Line 297: you mention paper from 2014, but there are many „more fresh“ papers, like:  Forgione, I., WoÅ‚oszyÅ„ska, M., Pacenza, M., Chiappetta, A., Greco, M., Araniti, F., ... & Bruno, L. (2019). Hypomethylated drm1 drm2 cmt3 mutant phenotype of Arabidopsis thaliana is related to auxin pathway impairment. Plant Science, 280, 383-396. They directly demonstrated link epigenetic, stomata behaviour and hormone auxin. It will be greta to mention at least in 1-2 sentences more recent epigenetic regulation of stomata development and it link with auxin.